# The Many Facets of Tumor Heterogeneity: Is Metabolism Lagging Behind?

**DOI:** 10.3390/cancers11101574

**Published:** 2019-10-16

**Authors:** Sara Loponte, Sara Lovisa, Angela K. Deem, Alessandro Carugo, Andrea Viale

**Affiliations:** 1Department of Genomic Medicine, The University of Texas MD Anderson Cancer Center, Houston, TX 77054, USA; SLoponte@mdanderson.org (S.L.); AKDeem@mdanderson.org (A.K.D.); 2Department of Cancer Biology, The University of Texas MD Anderson Cancer Center, Houston, TX 77054, USA; SLovisa@mdanderson.org; 3TRACTION platform, The University of Texas MD Anderson Cancer Center, Houston, TX 77054, USA

**Keywords:** intratumor heterogeneity, evolution, ecosystem, adaptation, complexity, genomics, epigenomics, reprogramming, metabolism, response to therapy, treatment resistance

## Abstract

Tumor functional heterogeneity has been recognized for decades, and technological advancements are fueling renewed interest in uncovering the cell-intrinsic and extrinsic factors that influence tumor development and therapeutic response. Intratumoral heterogeneity is now arguably one of the most-studied topics in tumor biology, leading to the discovery of new paradigms and reinterpretation of old ones, as we aim to understand the profound implications that genomic, epigenomic, and functional heterogeneity hold with regard to clinical outcomes. In spite of our improved understanding of the biological complexity of cancer, characterization of tumor metabolic heterogeneity has lagged behind, lost in a century-old controversy debating whether glycolysis or mitochondrial respiration is more influential. But is tumor metabolism really so simple? Here, we review historical and current views of intratumoral heterogeneity, with an emphasis on summarizing the emerging data that begin to illuminate just how vast the spectrum of metabolic strategies a tumor can employ may be, and what this means for how we might interpret other tumor characteristics, such as mutational landscape, contribution of microenvironmental influences, and treatment resistance.

## 1. The Discovery of Tumor Heterogeneity

Although optical microscopy dates back to the 17th century, it was not until 1833 that a young professor at Humboldt University of Berlin, Johannes Muller, would use this technology to differentiate tumors based on their architecture. Although better known for his contributions to anatomy and physiology, Muller was the first to describe, depict, and catalog epithelial and mesenchymal human tumors based on histological appearance [1,2,3]. Almost two centuries after publication, we can now appreciate that the tables and sketches Muller included in *“Cancer, and of Those Morbid Growths Which May Be Confounded with It”* already captured the heterogeneous essence of human tumors entirely (Figure 1). Muller’s pioneering work was further developed by his assistant, Rudolph Virchow, who published his landmark book, *Cellular Pathology*, 25 years later [4]. Virchow went on to be the first to document and describe an astonishing number of diseases and biological structures and processes, earning him the moniker, “*the Father of Modern Pathology*” or, to his peers, the “*Pope of Medicine*” [5]. Regarding tumors, Virchow’s detailed descriptions of pleomorphism among cancer cells and the tumor microenvironment confirm, like his mentor, that intra-tumor heterogeneity is a long-established clinical observation.

It soon became clear that cancer cells were heterogenous at the functional level as well. Although it had been known that only viable cells could propagate tumors from mouse to mouse in transplantation, it was not until the 1930s that it was demonstrated that not all viable tumor cells were endowed with the same tumorigenic capacity [6]. This was evaluated in humans in an ethically questionable study (which admits “most patients were interviewed before the procedure and informed of the experimental nature and general purpose of the study”) in which Southman and Brunschwig directly tested the autotransplantability of human cancer cells by subcutaneously injecting cell suspensions or tissue fragments derived from the patients’ own laparotomy or biopsy tissue to study the frequency of tumor development. Transplanted cells grew in less than 20% of patients studied, confirming the authors’ belief that not all cancer cells are equally capable of sustaining tumor growth [7]. These studies eventually led to the development of quantitative assays to investigate tumorigenic potential and, consequently, to the identification of cancer stem cells [8].

The realization that tumor cells are not all the same became even more compelling upon the demonstration in the late 1950–60s that tumors are monoclonal in origin. First, cytogenetic studies, then the assessment of allelic metabolic isoenzymes, immunoglobulin rearrangements, and genomic DNA polymorphisms, all consistently led to the same conclusion that human tumors derive from one single transformed cell [9]. To explain intrinsic variability among a clonal population of cells, it was hypothesized that tumors evolve and become progressively heterogenous over time. In 1976, only a few years after the identification of the Philadelphia chromosome, which was the first chromosomal rearrangement linked to the pathogenesis of a human neoplasia, Peter Nowell published his seminal work, “*The Clonal Evolution of Tumor Cell Populations”* [10]. Known as the “*Father of Tumor Evolution*”, Nowell conceptualized the first model of evolution, which is still considered valid. Based on published reports, he conceived tumorigenesis as a stepwise evolutionary process in which tumor cells acquire genomic alterations from time to time, that eventually generate new variants of altered cells. He writes, "*Nearly all of these variants are eliminated, because of metabolic disadvantage or immunologic destruction, but occasionally one has an additional selective advantage with respect to the original cells becoming the precursor of a new predominant subpopulation. Over time, there is sequential selection of sublines which are increasingly abnormal, both genetically and biologically.*" Of note, Nowell considered genomic instability to be the driving force of tumor evolution and was the first to recognize the environment and external perturbations, such as therapeutics, as important factors in shaping tumor heterogeneity [10].

A couple of years later, another fundamental manuscript was published by a group of scientists led by Gloria Heppner in which, through the characterization of distinct clonal lineages isolated from the same neoplasm, the authors formally demonstrated that cells with different genomic traits coexist within the same tumor and, more importantly, behave differently from each other [11]. This study represented an inflection point in cancer biology research and prompted the field to explore experimental methods to model intratumor heterogeneity and its influence on treatment response and drug resistance, invasion and metastasis, cross-talk among cells and between tumor cells, and the microenvironment [12,13,14,15,16,17,18]. Heppner was the first to introduce and experimentally test numerous concepts that are now tenets in cancer biology, including describing tumor spatial heterogeneity and postulating the existence of multiple mechanisms driving tumor evolution, including epigenetic alterations and changing microenvironmental stressors. However, Heppner’s name is likely best known for her population biology approach to the study of human tumors. She envisioned neoplasia as “*societies highly adapted for survival*”, where interactions between cells are critically important, and the equilibrium among different subpopulations of cells continuously change over time. In her words, “*the properties of the tumor cannot be deduced by the simple addition of its component parts … tumor societies survive natural and artificial (therapeutic) selection through heterogeneity by producing new variants to ‘outflank’ it”* [19]. Thus, Heppner was the first to recognize that interactions among clonal lineages influence the biological behaviors of tumors, including treatment response. Her vision and exceptional contributions to the field have been summarized in an essay published in 1984 [19], deservedly recognized as one of the most influential manuscripts ever published in *Cancer Research* [20].

## 2. Current Models of Tumor Evolution

Heppner’s definition of tumors as a “*Complex Ecosystem*” has increased in popularity due to new sophisticated technologies that have made it possible to validate many of her hypotheses. Over the last two decades, the advent and plummeting cost of next-generation sequencing (NGS), as well as the launch of cancer genomics programs led by non-profit and public consortia, have generated large-scale genomic datasets for tens of thousands of human tumors.

The first direct consequence of the genomic revolution is the revelation that long-standing models used to explain tumor progression and evolution were too simplistic. The step-wise “linear model” of tumor progression, first conceptualized by Foulds and Nowell [10,21] as a legacy of the old studies on chemical carcinogenesis and then expanded at the molecular level by Fearon and Vogelstein [22], indeed has a very limited application [23]. More current models address the accumulation and varied fitness of clonal lineages but differ in their description of how clonal variation emerges. In the *branching model*, tumor cells evolve and drift from a founder clone characterized by “trunk mutations” into branches with accumulating genetic diversity (“subclonal mutations”) [24]. This model, supported by multiple studies in numerous cancers, accounts for a continuous increase in tumor complexity wherein tumor lineages expand based on the fitness conferred by the constellation of newly acquired genomic abnormalities. An alternative model is the “*punctuate”,* or *“big bang model”*, which describes the acquisition of genomic aberrations as sudden, discrete mutational bursts or cataclysmic chromosomal events [23,25,26], such as chromothripsis [27]. These massive genomic events would occur early during tumorigenesis to create, at the onset of disease, the entire pool of clonal diversity that usually is found in late-stage tumors (Figure 2) [28,29].

Thus, as postulated by Heppner, it is now recognized that, despite a monoclonal origin, tumors at diagnosis are the variegated result of a complex process that yields many genomically, epigenetically, metabolically, and spatially different subclones that comprise a complex ecosystem (Figure 3). In fact, the progression of a tumor is shaped by its own heterogeneity, and different stages of tumor development maintain a structured cellular hierarchy that is phenotypically and spatially well-defined, and wherein less-aggressive clones spatially suppress their more aggressive counterparts [30]. Somewhat paradoxically, in the clinic, we must contend with the fact that anti-cancer drugs disrupt this equilibrium [31] and can even select for more aggressive clones in advanced, and more anaplastic stages of tumor evolution [32,33].

## 3. From Genomics to Metabolomics: The Role of Genes in Reprogramming Tumor Metabolism

It is well established that the genomic events responsible for transformation and tumor progression, such as activation of oncogenic signaling and loss of function of tumor suppressors, are also driving forces for the metabolic rewiring of tumors. Deregulation of glucose metabolism, first described by Otto Warburg in the 1920s and termed the *Warburg effect* in his honor, dominated the field of cancer metabolism for decades [34,35]. The “*aerobic fermentation*” described by Warburg (also known as aerobic glycolysis) [36], was based on the observation that tumors, unlike normal resting tissues, are able to produce lactate even in the presence of oxygen. Warburg attributed this event to the dysfunction of mitochondria in tumor cells that forced reliance on glycolysis for their energetics. However, although rapidly proliferating tumor cells are highly glycolytic and can use lactate production to regenerate NAD+, it is now understood that the upregulation of glucose and glutamine consumption by dividing cells is to accommodate the molecular building blocks needed for biosynthetic purposes as opposed to increased energy requirements [37]. Moreover, the idea that mitochondrial defects are the primary driver of glycolytic flux has been discarded due to the demonstration that aberrant activation of prototypical oncogenes, such as *MYC*, *KRAS*, *EGFR*, *PI3K*, *AKT*, profoundly affects cell metabolism and increases the uptake and utilization of glucose and glutamine (Figure 4) [38,39,40,41,42,43].

Another key mediator of glycolytic metabolism in tumors is HIF1α, which belongs to the family of hypoxia-inducible transcription factors. HIF1α is a central player in oxygen sensing and homeostasis [44]. The transcriptional activity of HIF complexes is suppressed during normoxia; however, when molecular oxygen is insufficient to support normal dioxygenase activity (e.g., hypoxia), HIF1α is not hydroxylated at prolyl and asparaginyl residues. As a consequence, in the absence of hydroxylation, HIF1α is not recognized as a substrate for proteasomal degradation by the ubiquitin ligase pVHL–elonginB–elonginC complex, and the increased level of HIF induces complex transcriptional programs that sustain angiogenesis, cell migration, and proliferation, as well as glycolytic activation [44]. Because of the hypoxic microenvironment caused by fast-expanding masses, HIF activation is a common feature of many human cancers. In addition to hypoxia, HIF stabilization and activation of downstream transcriptional programs can result from loss of function mutations of *VHL*, as well as mutant alleles of metabolic enzymes, such as *FH-* or *SDH-* mutant tumors, in which high levels of the TCA intermediates fumarate or succinate, respectively, interfere with dioxygenase activity and increase HIF1α stability [43,44]. These facts suggest that multiple oncogenes and transformational events all lead to the same phenotypic outcome: activation of a common set of metabolic programs that increase glycolytic flux. But, should we expect this to be the case?

To address this important issue, we must first consider that what has been described as tumor metabolic reprogramming or rewiring is, in reality, not a feature specific to tumor cells. In fact, tumor metabolism, including the Warburg effect, recapitulates the metabolism of actively dividing normal cells [45]. To undergo a division and generate two daughter cells, both normal and cancer cells rely on activation of the same biosynthetic programs to expand biomass, and because the major carbon sources that fuel the increased anabolic processes are glucose and glutamine, all dividing cells rely on glycolysis and glutaminolysis [38,39,40,41,42,43]. Glycolysis, the breakdown of one six-carbon molecule of glucose into two three-carbon pyruvate molecules, is probably the most important metabolic pathway for dividing cells. The intermediate molecules of glycolysis fuel multiple collateral anabolic pathways, making glycolysis the hallmark of active proliferation. Glycolic metabolites fuel the generation of nucleotides (ribose), triglycerides, phospholipids (glycerol), and important amino acids such as alanine, serine, and glycine, and they provide reducing equivalents for anabolic reactions (NADPH). Pyruvate, the final product of glycolysis, if not converted into lactic acid by lactate dehydrogenase (LDH), enters the citric acid cycle (TCA) as acetyl-CoA or oxaloacetate, where pyruvate-derived carbo-skeletons can be used as intermediates for other biosynthetic processes, such as synthesis of fatty acids or cholesterol. Like glucose, glutamine is an important source of carbon and nitrogen for dividing cells [40,46]. Upon uptake, glutamine is converted to glutamate by glutaminase (GLS), and subsequently to α-ketoglutarate after modification by transaminases (GOT) or glutamate dehydrogenase (GLDH). α-ketoglutarate enters the TCA cycle and, through further modifications to oxaloacetate, sustains the generation of aspartate, an essential substrate for nucleotide synthesis. Glutamine and glutamate also serve as key nitrogen donors for many transamination reactions important for the production of other non-essential amino acids [46]. In light of this heavy reliance on glucose and glutamine to supply molecular intermediates toward the synthesis of all four major types of biomolecules, it becomes clear why cells increase glucose and glutamine uptake to divide.

The coordination of the cell cycle with changes in anabolic metabolism during cell division is largely through the *MYC* family of transcription factors (hereafter *MYC* refers to *cMYC*). Thought to be a general transcriptional amplifier that targets all active promoters and enhancers in the genome [47], it has recently been demonstrated that *MYC* regulates a discrete set of genes [48]. A critical node downstream of distinct signaling pathways that lead to cell growth and division, MYC executes its proliferation program also through the activation of metabolic functions that fulfill the anabolic requirements of a dividing cell, including genes that control nucleotide and RNA metabolism, ribosome biogenesis, protein synthesis, and energetic (glucose) metabolism [39,48]. Beyond MYC, a direct link between the Warburg effect and the cell cycle machinery has also been documented, which lends additional support to an intrinsic coupling between the cell cycle and anabolic metabolism [49]. It has been demonstrated that, in normal dividing cells, such as embryonic cells or T-lymphocytes, the anaphase-promoting complex/cyclosome-Cdh1 (APC/C-Cdh1), a key regulator of the G1-S transition, inhibits glycolysis and glutaminolysis. Through its E3 ligase activity, the APC/C-Cdh1 complex targets 6-phosphofructo-2-kinase/fructose-2,6-bisphosphatase 3 and glutaminase-1 for degradation. Because the APC/C-Cdh1 complex is tightly regulated during the cell cycle, its inactivation at the initiation of S-phase would enhance glycolytic flux and glutaminolysis. Reactivation of the complex in late mitosis, when the biosynthetic needs of cells decrease, would reverse this effect [50,51]. As motifs recognized by Cdh1 have been identified in many other metabolic enzymes, including pyruvate carboxylase, malate dehydrogenase-1, and acetyl-CoA carboxylase-1, one can speculate that the coupling between cell cycle machinery and metabolism may exert a broader role in the regulation of anabolic processes during specific phases of cell division [51].

Not surprisingly, tumor suppressors exert exactly opposite effects of MYC on cell metabolism, suggesting a “gatekeeper” role for metabolism programs to restrain proliferation and perhaps transformation. TP53, through transcriptional regulation of target genes such as TP53-induced glycolysis and apoptosis regulator (*TIGAR*) and *Parkin*, down-regulates the glycolytic pathway [52,53]. TIGAR, a fructose bisphosphatase, drastically reduces glycolytic flux by degrading fructose-2,6-bisphosphate, a potent positive allosteric effector of 6-phosphofructo-1-kinase (PFK1) [52]. p53 also suppresses glycolysis through the direct transcriptional repression of glucose transporters, hexokinases, and phosphoglycerate mutase enzymes [54,55,56,57]. The regulation of mitochondrial activity by p53 has also been described. Through the downregulation of the lactate/proton symporter, monocarboxylate transporter 1 (*MCT1*), as well as downregulating enzymes that inhibit pyruvate dehydrogenase (PDH) activity, p53 activity directly limits lactate production and favors the conversion of pyruvate into acetyl-CoA to fuel the TCA cycle [58,59]. p53 also regulates mitochondria biogenesis [60,61] and the expression of proteins involved in the assembly and maintenance of respiratory complexes in the electron transport chain [62,63]. All of these metabolic regulations are lost upon mutation or deletion of *TP53* [64,65].

If all proliferative programs lead to anabolic metabolism, it might be expected that, in comparison with other cellular phenotypes, metabolism programs should be relatively homogeneous among tumors. However, we are now keenly aware that this is not the case. Although all oncogenes drive proliferation, distinct oncogenes activating different signaling networks can variably engage metabolic pathways, resulting in differential utilization of metabolites [46,66,67]. Also, tumors arising from different tissues, even if they share the same driver mutations, may present with distinct metabolism due to the differences in the cell of origin [43,67]. Further, external perturbations to the tissue of origin or the tumor may influence cell metabolism. Thus, despite their common disposition to proliferate and spread, tumors can dysregulate and hijack metabolic programs in numerous, highly specific ways to survive.

## 4. Metabolic Intratumor Heterogeneity

Because genomic events profoundly affect tumor metabolism, based on the remarkable genomic and functional heterogeneity described within a single cancer, should we expect a corresponding degree of intratumor metabolic heterogeneity? Driver mutations usually represent ‘*trunk*’ genomic events—aberrations acquired early during progression and shared by the vast majority of cells within the tumor—which may suggest that little or no functional variability would be expected between tumor cells. However, considering cell proliferation as an excellent example of a tumor cell phenotype, we recognize striking variability among cells that share a similar mutational burden, wherein dividing cells are typically represented by a very small subpopulation. This was first demonstrated in the late 1960s when Bayard Clarkson conducted pulse-chase experiments with 3H-thymidine directly in patients with acute myeloid leukemia (AML) to evaluate tumor proliferation in vivo [68,69]. His unbiased approach demonstrated that, even in this extremely aggressive form of cancer, leukemic blasts were almost entirely post-mitotic, with only a minority (5%) of cells actively dividing. His work went on to show that the proliferating cell subpopulation was not homogeneous, but was comprised largely of fast-cycling cells (doubling time of one day), as well as a smaller proportion of slow-growing cells (termed “*dormant*”) that were characterized by infrequent divisions (doubling time lasting from weeks to months). Similarly, using molecular barcoding to track the evolution of primary human tumors in vitro and in vivo, our group recently found that, in pancreatic cancer, ~70% to 80% of cells are post-mitotic and functionally exhausted [31].

The clear evidence that tumor cells can behave so differently, even being mutationally similar and possessing a potent oncogene such as mutated KRAS, reminds us again of Heppner’s description of tumors as complex ecosystems. We know now that tumors, like normal tissues, are organized in hierarchies consisting of slow-growing stem cells, transiently amplifying progenitors, and a large number of cells that have exited the cell cycle [70]. Our explosion in understanding epigenetic regulation at the molecular level in developmental and oncogenic programs has synergized to shed new light on the intersection between tumor hierarchy and genetic context. In brief, different epigenetic states or degrees of differentiation characterize the functional compartments of a tumor and, apparently, can override oncogenic signaling. Indeed, the very same oncogene can lead to distinct outputs (cycling vs. non-cycling) when cells transition from one chromatin state to another. Several elegant studies have demonstrated this directly, showing that both tumor cells and nuclei can be successfully reprogrammed to induced-pluripotent or embryonic stem cells irrespectively of their mutational burden. These undifferentiated cells are able to differentiate back to apparently ‘normal’ cells, and even contribute to the development of adult mice when injected into blastocysts [71,72,73]. Although extreme examples, these data strongly suggest that epigenetic context and chromatin organization can modulate the oncogenic activity of genomic events.

Similarly, there is direct evidence that epigenetic states can dictate metabolic programs to generate functionally heterogeneous subpopulations of cells that share a nearly identical mutational signature. In pancreatic cancer, our group uncovered the spontaneous emergence of mesenchymal lineages upon the dysregulation of the chromatin remodeling complex SWI/SNF [74]. Although these mesenchymal cells harbored the same oncogenic *KRAS* mutation as their more epithelial counterparts, these cells were highly aggressive and characterized by low engagement of MAPK signaling and robust activation of MYC that induces protein anabolism, biomass accumulation, and adaptive response to stress [74]. Since the publication of this manuscript, continued work to characterize the metabolism of clonal lineages has uncovered chromatin modifications that are likely regulating the differential metabolic programs in tumor cells bearing the same genomic aberrations. Another group has recently described a link between SWI/SNF and tumor cell metabolism, reporting a shift toward oxidative metabolism in lung tumor cells with SMARCA4 loss or mutation [75]. In melanoma, cells with high expression of KDM5B (JARID1A), a histone demethylase, have deregulated bioenergetics and are characterized by a sustained up-regulation of proteins involved in the electron transport chain, the multiprotein enzymatic complexes responsible for mitochondrial respiration, as well as a significant down-regulation of glycolytic enzymes [76,77]. Differential expression of KDM5B can thus be used to identify a small subpopulation of melanoma cells characterized by distinct bioenergetics that are resistant to various drugs.

Today it is well accepted that tumors rely on mitochondrial respiration, and there is a growing list of studies that identify specific cell subpopulations characterized by a more oxidative metabolism with respect to other tumor cells. In particular, slow-growing cells, dormant cells, and cancer stem cells are generally highly reliant on oxidative phosphorylation [34,78]. In pancreatic cancer, we identified a subpopulation of cells that were not addicted to the oncogenic signaling and that were dependent on oxidative metabolism for their survival [79]. Although we first hypothesized that the shift to oxidative metabolism might reflect the low energetic and anabolic rates of these quiescent cells, our recent studies support a much more complicated scenario. Using an innovative platform to generate large cohorts of patient-derived xenotransplants, in which all animals bear tumors with identical clonal composition or *Clonal Replica Tumors (CRTs),* we studied heterogeneous populations of cells and their clonal dynamics in vivo in response to multiple pharmacological perturbations [31]. Through a systematic and quantitative evaluation of the effects of single- and combined-therapies on tumor clonal composition, we uncovered that relapsed pancreatic tumors previously treated with drugs with unrelated mechanisms of action (gemcitabine, MEK, and PI3K inhibitors) have completely different clonal architecture. This strongly suggests that, inside the same human tumor, different populations of cells characterized by differential sensitivity to drugs preexisted and acquired increased fitness upon therapeutic challenge. Surprisingly, when we isolated and deeply characterized twelve treatment-naïve clonal lineages based on their in vivo differential sensitivities to drug treatment, we uncovered astonishing heterogeneity with regard to deregulated metabolic pathways. Altered regulation in nearly every metabolic pathway, including oxidative phosphorylation as well as folate, fatty acid, essential amino acids, and sugar anabolism and catabolism, were all observed across the different lineages [31].

Although we do not yet know the full extent or significance of this unexpected variability among subclonal lineages that populate the same tumor, these data strongly suggest that metabolic heterogeneity exists and is a pervasive feature of human tumors, at least in pancreatic cancer. Our ability to characterize, capture, and describe the spectrum of metabolic programs present within a tumor has been limited due to the low sensitivity of the technical approaches used to investigate metabolism. Traditional metabolomics and isotope tracing are excellent tools to investigate the average contribution of metabolites to all of the cells under study, but these techniques cannot assess the heterogeneous spectrum of metabolic programs that is being reflected in those averages. Major technological advancements, such as the further development of single-cell metabolomics mimicking the evolution of genomics and transcriptomics, will be required to precisely define the metabolic landscape of tumors. Currently, mass spectrometry imaging (MSI) is the most advanced way to explore metabolic heterogeneity, and both matrix-assisted laser desorption ionization (MALDI) and secondary ion mass spectrometry (SIMS), which already boast sub-micron resolution, will play an important role in the further development of this technology [80].

Regional differences in the vasculature, stromal architecture, cell density, and viability contribute to intratumor spatial variegation that can be identified by positron emission tomography (PET), CT, and MRI, especially when used in combination with contrast dyes in order to reveal dynamic changes in blood perfusion. Although these imaging technologies in common use in clinical oncology enable the detection of structural and, to some extent, functional tumor heterogeneity, the data collected is not effectively used in clinical practice [81]. Because radiologic images represent more than simple pictures [82], in recent years, the rapid expansion of computational modeling and analysis capabilities has led to the development of ‘radiomics’, which are new approaches to extract, quantitatively analyze, and mine all data from digital images deposited in large shared databases, with the goal to improve clinical outcomes for patients with cancer [82,83]. This includes also efforts made by multiple investigational groups to develop new, highly sensitive technologies to investigate metabolism in vivo, such as hyperpolarized MRI [84,85], or to integrate imaging technologies with genomics and molecular profiling, which have led to substantial advances in our ability to interpret complex radiological images in specific disease contexts [81,86,87,88]. One notable example of the application of this multipronged approach is the recent work of the research group led by Ralph DeBerardinis. Building on previous studies assessing the feasibility of isotope tracing in vivo [89,90,91], the authors demonstrated the existence of regional metabolic heterogeneity in patients affected by non-small-cell lung cancer [92]. After the evaluation of glucose uptake, cellularity, and tumor perfusion of pulmonary lesions through ^18^fluoro-2-deoxyglucose positron emission tomography (FDG-PET) and multi-parametric MRI, patients received an intraoperative, continuous infusion of unilabelled-^13^C-glucose at the time of surgery. After explant, tumors were subjected to histological and molecular analysis as well as metabolomics and NMR to trace the contribution of glucose-derived carbons to tumor metabolism. This comprehensive approach enabled the authors to determine that glucose utilization, while increased in tumors compared to normal tissue, varies within the same lesion and is affected by regional perfusion. Specifically, low-perfused tumor regions were characterized by a more sustained contribution of glucose to central metabolism versus high-perfused regions, suggesting that tumor cells in less vascularized areas rely more on oxidative metabolism. These observations were consistent with transcriptomic analyses indicating that low-perfused tumors were enriched for gene pathways such as glycolysis, mitochondrial respiration, and the TCA cycle, whereas high-perfused tumors were enriched in the lysosome and amino acid metabolism [92]. These findings suggest that lung tumors sustain mitochondrial metabolism through a variety of substrates besides glucose, including lactate [93]. In addition to providing an elegant demonstration of the metabolic heterogeneity of human tumors, these observations demonstrate that the enhanced uptake of ^18^FDG-PET in vivo is not an index of tumor glycolytic addiction; rather, it is the consequence of increased glucose oxidative metabolism. These pioneering studies represent a direct challenge to assumptions regarding the Warburg effect in tumor cells that have shaped decades of research into tumor metabolic programming and encourage continued debate regarding current and historical approaches to investigate cellular metabolism in the lab [94].

## 5. Metabolic Heterogeneity in the Tumor Microenvironment

Tumors are complex tissues comprised of a variety of different cells. As a consequence, besides their intrinsic metabolic requirements, the modulation of cancer cell metabolism can be extrinsically influenced through interactions with the tumor microenvironment. The complex and intricate network formed by cancer cells, matrix-depositing proliferating cancer associated-fibroblasts (CAFs), dysfunctional blood vessels, and immune cells dramatically influence tumor architecture. Inside the tumor, cells must contend with hypoxia, increased interstitial pressure, augmented stiffness, and nutrient deprivation, all of which profoundly shape the metabolic requirements of both stromal and cancer cells [95]. Amongst cells in the tumor, heterogeneity emerges as cells tune their metabolism programs to compensate for the adverse microenvironment, where nutrient competition leads to the repurposing of nutrients and metabolites, and cells adopt intrinsic metabolic strategies as well as extrinsic nutrient sharing mechanisms to support growth and survival. Below, we review the most recent findings regarding the contribution of two major microenvironmental constituents, CAFs and endothelial cells, to shaping tumor metabolism.

### 5.1. Cancer Associated-Fibroblasts (CAFs)

CAFs participate in the host response to tissue injury caused by cancer cells. They have been shown to critically impact tumorigenesis, and they constitute a synthetic machine that produces many different tumor components [96]. In addition to paracrine signaling, which is the focus of the following paragraphs, direct cellular contacts between CAFs and cancer cells influence metabolic programs [97]. Additionally, symbiotic metabolic reprograming between CAFs and cancer cells can also be mediated by the release of small extracellular particles, such as exosomes [98,99,100]. Prostate and pancreas CAF-derived exosomes supply cancer cells with a plethora of metabolites, including amino acids, lipids, and TCA-cycle intermediates [98,99]. Conversely, breast cancer-derived exosomes can activate MYC signaling in CAFs through exosome-encapsulated miR105, to induce metabolic reprograming of stromal cells [100].

Multiple lines of evidence support that the pro-tumorigenic role of CAFs is exerted by influencing the metabolic microenvironment of tumors. Cancer cells can highjack CAFs to produce energy-rich metabolites to burn via oxidative phosphorylation, in a process termed the *reverse Warburg effect* [101]. Cancer cells induce oxidative stress in CAFs by producing reactive oxygen species (ROS), which upregulates HIF1α expression in neighboring CAFs due to the inhibition of PHD proteins. In CAFs, increased HIF1α promotes autophagy and degradation of caveolin-1, which negatively regulates nitric oxide (NO) production. The consequent excessive production of NO leads to mitochondria dysfunction, further increases in ROS, mitophagy, and upregulation of glycolysis [102]. As a consequence, CAFs become highly dependent on aerobic glycolysis and, therefore, produce large quantities of lactate that is then shuttled to cancer cells by the monocarboxylate transporters, MCT4 and MCT1, on CAFs and cancer cells, respectively [97,103]. TGFβ and PDGF signaling pathways can also induce a HIF1α-mediated metabolic switch towards glycolysis in CAFs. The stabilization of HIF1α under normoxic conditions is reached by a downregulation of the isocitrate dehydrogenase 3 complex (IDH3α). This decreases the intracellular levels of α-ketoglutarate, which, in turn, stabilizes HIF1α by preventing its PHD2-mediated degradation [104]. Similarly, ketone bodies, such as 3-hydroxy-butyrate, are also generated as end-products of aerobic glycolysis in CAFs and then utilized by cancer cells to fuel oxidative phosphorylation and anabolic metabolism [105,106,107]. This metabolic symbiosis between CAFs and cancer cells, however, does not seem to apply to all CAF-cancer cell interactions. Studies have essentially documented the opposite phenomenon in ovarian and pancreatic cancer, wherein lactate is released in the microenvironment by glycolytic cancer cells and used by CAFs with low glycolytic activity to fuel oxidative phosphorylation [108,109].

Numerous studies also point to an important relationship between CAFs and cancer cells with regard to amino acids. Alanine secreted by autophagic pancreatic stellate cells fuels the TCA cycle of cancer cells and supports biosynthesis of lipids and non-essential amino acids, therefore serving as an alternative carbon source that allows cancer cells to bypass the drastic nutrient depletion in the pancreatic tumor microenvironment [108]. In ovarian tumors, where glutamine is rare, CAFs upregulate glutamine anabolic pathways to enable glutamine synthesis from atypical sources, such as glutamate and lactate secreted by ovarian cancer cells. Consistent with these observations, co-targeting glutamine synthase and glutaminase impairs ovarian tumor growth and metastasis, representing a promising synthetic lethal approach to target tumor-stroma interdependent metabolism [109]. Stromal glutamine has also been detected in glioblastoma, where astrocytes can provide glutamine to brain cancer cells [110], and in pancreas cancer, where adipocytes secrete glutamine that sustains pancreas cancer cell growth [111]. Stromal release of cysteine has been associated with resistance to chemotherapy in both chronic lymphoid leukemia and ovarian cancer [112,113]. Also in ovarian cancer, cytotoxic CD8^+^ T cell activity impairs cysteine and glutathione release by CAFs, thereby synthetizing ovarian cancer cells to chemotherapy [113]. Conversely, the deprivation of amino acids due to competition among cells in the tumor microenvironment can generate an immune-suppressive microenvironment that strongly inhibits the capacity of immune cells to kill cancer cells. Tryptophan catabolism by elevated IDO1 in CAFs generates immunosuppressive kynurenine metabolites, resulting in T cell anergy and apoptosis and negative regulation of dendritic cell immunogenicity [114,115,116]. Similarly, the secretion of Arginase 2 by CAFs can deplete arginine in the tumor microenvironment and impair T cell proliferation and function [117,118,119]. Nutrient competition, however, does not only involve the subtraction of these two amino acids from the microenvironment but can also involve glucose deprivation by high-consuming cancer cells, with consequent glucose deprivation of antitumor effector T cells [120,121].

### 5.2. Endothelial Cells

Within a tumor, the increasing demand for oxygen, combined with the establishment of regional hypoxia, strongly drives the formation of new blood vessels from pre-existing ones through the tightly regulated process of angiogenesis [122]. Energy-rich metabolites present in the tumor microenvironment are among the stimuli that can trigger angiogenesis. For example, lactate produced by cancer cells can be uptaken directly by endothelial cells through the MCT1 transporter and stimulate tumor angiogenesis by fueling the TCA cycle and inducing an autocrine NF-κB/IL-8 pathway [123,124]. In addition, lactate acts as a proangiogenic factor by binding and stabilizing NDRG3 (N-MYC downstream-regulated gene 3), which in turn triggers angiogenic signals during hypoxia [125].

The metabolic profile of tumor-associated endothelial cells has not yet been fully elucidated; however, endothelial cell metabolism has recently become a subject of intense investigation. Multiple studies have led to the discovery that the transition of endothelial cells from a quiescent to an angiogenic profile is accompanied by a defined metabolic switch that plays a fundamental role in regulating endothelial cell function during angiogenesis. Endothelial cells are highly dependent on glycolysis rather than oxidative phosphorylation for their ATP production, and further upregulation of glycolysis occurs during angiogenesis [126]. The preference by endothelial cells for aerobic glycolysis seems counter-intuitive given their direct exposure to blood and oxygen, but this phenomenon has been explained by various mechanisms: (1) by consuming less oxygen, endothelial cells leave more oxygen available to transfer into the tissue; (2) angiogenesis intrinsically requires the formation of new blood vessels in non-vascularized tissues, which is best supported by cells that can rely on anaerobic glycolysis; (3) energy production by glycolysis is faster than oxidative phosphorylation; and (4) by limiting oxidative phosphorylation, endothelial cells are also limiting the formation of ROS [126].

The glycolytic activator PFKFB3 was shown to regulate not only endothelial cell proliferation but also cytoskeletal rearrangements required for vessel sprouting, and the inhibition of PFKFB3 was shown to reduce pathological ocular angiogenesis and normalize tumor vessels, improving chemotherapy and inhibiting metastasis [126,127,128]. Oxidation of fatty acids (FAO) has also recently been reported to sustain the proliferation of endothelial cells involved in vessel sprouting by providing a source of carbon for de novo synthesis of nucleotides for DNA replication [129]. Genetic and pharmacological inhibition of CPT1A, an FAO rate-limiting acyl-CoA mitochondrial transporter, has been shown to improve pathological ocular angiogenesis by the proliferation of endothelial cells [129], but further studies will be needed to elucidate whether it is conceivable to apply this therapeutic strategy to target tumor angiogenesis. Interestingly, quiescent endothelial cells present 3-fold higher FAO compared to proliferating cells, and FAO has a protective role in sustaining the TCA cycle for redox homeostasis through NADPH regeneration [130]. Inhibition of FAO induces a dysfunctional endothelial cell phenotype [130] and a pathological mesenchymal profile [131,132], suggesting that FAO is required for the maintenance of endothelial cell identity.

Recent investigations have uncovered important contributions of amino acids, particularly glutamine, in the regulation of endothelial cell metabolism and function. Glutamine metabolism is required for endothelial cell proliferation [133,134], and although the specific role of glutamine in regulating tumor angiogenesis has not yet been characterized, depletion of glutamine levels in the tumor microenvironment through inhibition of glutamine synthase in tumor-associated macrophages resulted in tumor vessel normalization and decreased metastasis [135]. Glutamine synthase has also been recently described to exert a role in endothelial cell migration by regulating Rho GTPases during pathological angiogenesis [136]. Although our comprehension of the metabolic changes associated with endothelial cells and angiogenesis during tumor development is still in its infancy, all the emerging work clearly indicates that targeting the metabolism of endothelial cells may represent a promising therapeutic strategy to inhibit tumor angiogenesis.

## 6. Metabolic Heterogeneity of Cancer Treatment Responses

Differences in the metabolic flux between normal and tumor cells represent dependencies that can be exploited to specifically target cancer cells [38]. Metabolic rewiring and redox balance maintenance constitute the result of finely tuned and interconnected molecular and metabolic pathways that have been selected to bestow human cells with the ability to react and adapt to external changes quickly. Tumors exploit this metabolic flexibility to escape environmental pressures, including drug treatment, which has recently been demonstrated by the inefficiency of cancer metabolism therapies [137,138]. Indeed, cancer cells have proven to be extremely plastic with regard to metabolism, and during tumor progression or in response to stressors, they are able to switch between alternative metabolic phenotypes, like glycolysis and oxidative phosphorylation [40].

These distinct metabolic states of glycolytic versus oxidative are governed and coordinated by master regulators, such as AMP-activated protein kinase (AMPK) and HIF-1, which act as molecular rheostats guiding intracellular adaptations to external perturbations [139]. However, thanks to experimentally validated model predictions, it has recently been demonstrated that cancer cells have the ability to adopt additional metabolic states not typical of normal cells, challenging the conventional dichotomous classification of tumor metabolism [140]. To manage this complexity, several frameworks have been constructed that aim to reduce the size of an extensive regulatory circuit to basic components, and yet capture its fundamental principles and overall network behavior [138,141,142]. Leveraging such metabolic network-deconvolution strategies has identified multiple genes/proteins that can predict cancer drug response or resistance [143,144,145,146]. For instance, as a central carbon source for the cell, glucose metabolism is highly articulated. Many enzymes contribute to the series of reactions necessary to catabolize glucose, and some key components of the glycolytic pathway, such as glucose transporters (GLUTs), hexokinase (HK), pyruvate kinase M2 (PKM2), and LDHA, have been selectively exploited to enable glycolytic inhibition as an anticancer strategy [147,148,149,150,151]. Similarly, dichloroacetate is a pyruvate dehydrogenase kinase (PDK) inhibitor that is being explored clinically with promising results [152,153]. By inhibiting PDK and, subsequently, pyruvate dehydrogenase (PDH), the rate-limiting step between glycolysis and the TCA cycle can be targeted to flip a metabolic switch from glycolysis to mitochondrial respiration and induce apoptosis.

Two additional key metabolic pathways that have been extensively dissected to identify optimal points of intervention are the fatty acid biosynthesis pathway, which synthesizes lipid synthesis from basic metabolites like acetyl-CoA and malonyl-CoA, and glutaminolysis, which regulates cell growth and energy metabolism by converting glutamine to α-ketoglutarate (α-KG). To target fatty acid biosynthesis, fatty acid synthase (FASN) has emerged as a promising anticancer target. FASN inhibitors may sensitize cells to chemotherapy or enhance the efficacy of other targeted therapies [154,155], and several FASN inhibitors have shown single-agent antitumor activity [156,157]. Small molecule inhibitors of glutaminase (GLS) effectively shut down glutaminolysis, which plays a critical role in tumor cell metabolism, and several GLS inhibitors are in various stages of pre-clinical or clinical development [46,158,159,160].

To maximize the clinical impact of the growing list of metabolic inhibitors in oncology practice, it is essential that we continue to expand our understanding of tumor metabolic profiles, heterogeneity, and adaptive response [143,161]. In some cases, the presence of dominant drivers or tumor suppressor mutations that dysregulate metabolic pathways provides useful context to select treatment. For example, the antitumor activity of caloric restriction appears to be entirely abrogated by the activation of PI3K or inactivation of PTEN [162]. Similarly, while dietary restriction of glucose can inhibit the growth of many cancers, in some contexts, such as in the absence of an isoform of protein kinase C or in the presence of certain mutant p53 alleles, glucose restriction can paradoxically lead to more aggressive tumors [163,164]. However, as discussed above, there exists a vast spectrum of specific metabolic programs both among tumors and within a single tumor. How, then, can specific metabolic programs be identified to uncover therapeutically relevant dependencies? PET imaging can measure glucose uptake by tumor cells, but it does not output functional information regarding the utilization of metabolic pathways inside the cells [165]. Other noninvasive approaches, such as magnetic resonance spectroscopy (MRS), which can measure metabolite concentrations in tumors, combined with metabolomic profiling of serum or urine, may produce a data package that can guide the selection of appropriate targeted therapy(ies) in a tumor- or patient-specific fashion [166,167]. Additionally, tools such as nanoproteomic assays could be developed to monitor target gene expression and quantify signaling in rare tumor cell populations, and such has already been accomplished for AKT1/2/3 and 4EBP1 in acute myeloid leukemia cells [168].

Both pre-clinical and clinical studies are clear that even effective targeted metabolic therapies cannot produce durable disease remissions or cures, and understanding how cancer cells rewire their metabolism under pressure—how “compensatory” metabolism negatively impacts treatment responses—is an area of intense investigation [145]. As a very direct example of this, the use of mitochondrial inhibitors such as oligomycin induces a glycolytic phenotype, whereas glycolytic inhibitors enhance the activity of AMPK and induce an oxidative phenotype [169]. This metabolic plasticity can be thwarted with dual inhibition of both glycolytic and mitochondrial respiration, as clearly exemplified by a recent study in which the authors elegantly demonstrated that metformin plus fasting-induced hypoglycemia synergistically impacts tumor growth [170]. Other recent studies have shown that a subset of BRAF-mutated melanoma cells that are resistant to BRAF inhibitors can activate the MITF-driven expression of PGC1a to upregulate mitochondrial respiration to evade therapy [171]. This phenomenon has been observed upon MEK or PI3K inhibitor treatment in oncogenic KRAS-driven tumors, such as pancreatic cancer, as well [79]. Moreover, there are data to support that the effects of BRAF inhibition are maximized when melanoma cells are heavily reliant on glycolysis and/or when depletion of mitochondria forces cells to solely utilize glycolysis [172]. Together, these studies suggest identifying and targeting the major compensatory metabolic pathways induced upon drug treatment can enhance therapeutic benefits by illuminating rational drug combinations. Such approaches are also certain to contribute to our ability to profile patient tumors in real-time to deliver personalized therapeutic regimens.

The effectiveness of metabolic or cytotoxic therapies is strongly influenced by spatial heterogeneity with regard to both the tumor structure/microenvironment and drug diffusion [173]. For example, selective intratumoral lethality, or intra-tumor metabolic zonation, is a recently described phenomenon in which a drug selectively kills less aggressive clones while sparing more malignant populations based on their relative distance from blood vessels, and it has been confirmed to affect antitumor drug effects in a majority of solid tumors [174,175]. Similarly, acidification of intratumoral regions by cytotoxic agents or due to proximity to areas where extensive cell death has occurred can mediate invasion by metabolically distinct neighbor clones and impose selective pressure to define tumor evolution [176,177].

## 7. Concluding Remarks

Our knowledge of the mechanisms by which phenotypic, temporal, spatial, and molecular heterogeneity influence tumor growth and drug response, represent decades of attempts to deliver pharmacological cures to patients with cancer. It is clear that one of the greatest challenges in current oncology practice is to develop methods to characterize and exploit the metabolic and other cancer-specific programs that endow tumors with the ability to adapt to and evade therapeutic assault. By understanding the highly heterogeneous and plastic microenvironment of tumors, and through continued characterization of oncogenic reprogramming of cancer cells across tumor types, we can aim toward the development of biomarkers and techniques that can support the real-time analysis of clinical samples to design personalized therapeutic drug regimens. Undoubtedly, targeting the multitude of potential metabolic strategies that tumors can activate will play an essential role in these efforts as the field continues to push forward.

## Figures and Tables

**Figure 1 cancers-11-01574-f001:**
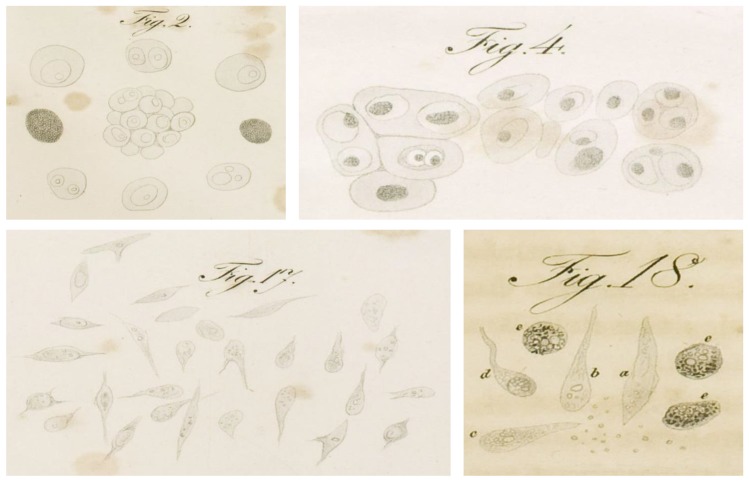
Reproduction of original drawings representing the microscopic appearance of cancer cells isolated from different human neoplasia, as reported in Muller’s 1838 work on cancer [1]. Panel 2 and panel 17 depict mono- and poly-nucleated tumor cells in a “reticular” carcinoma, and heterogeneous spindle-shaped cells isolated from a lower jaw osteocarcinoma respectively (Table II, page 69); panel 4 represents different polynucleated cells isolated from a tumor of the parotid gland (Table III, page 71); panel 18 shows different morphological cells comprehending pigmented cells (**e**) isolated from an osteocarcinoma (Table I, page 67).

**Figure 2 cancers-11-01574-f002:**
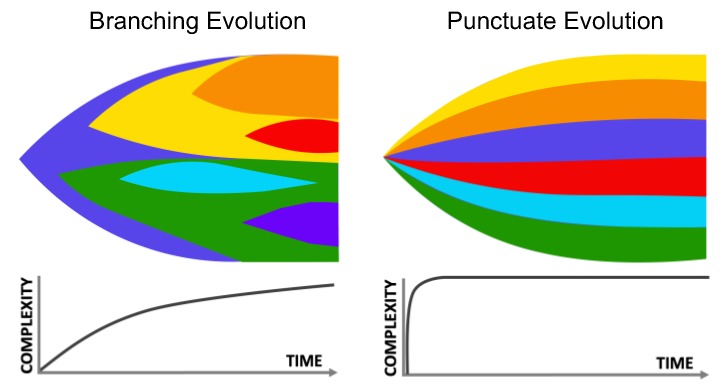
Current models of tumor evolution. According to the branching model (left), tumor complexity increases over time due to continuous accumulation of genomic events; in a punctuate evolution model (right), the full tumor complexity is acquired early during progression due to cataclysmic genomic events.

**Figure 3 cancers-11-01574-f003:**
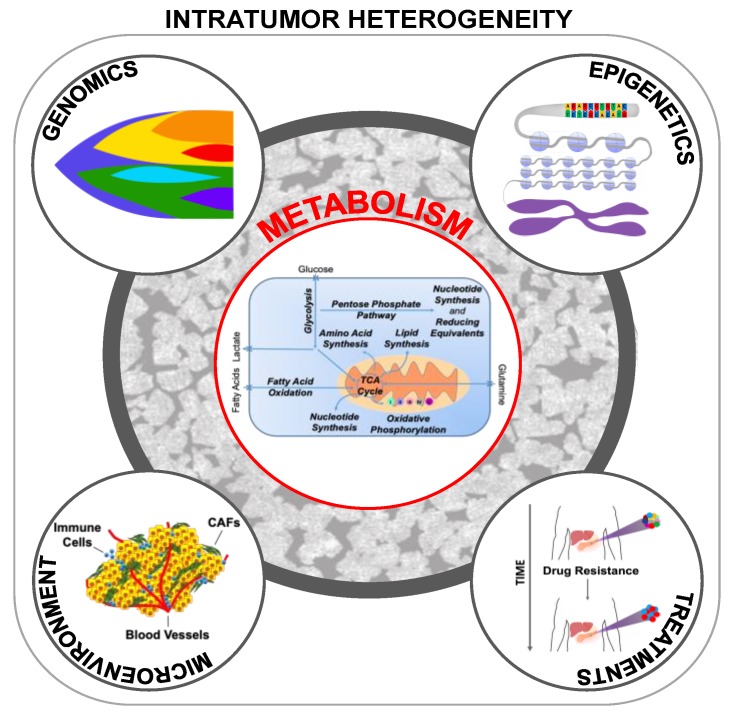
Tumors are complex ecosystems that constantly evolve over time in response to intrinsic and extrinsic perturbation. The principal components of tumor heterogeneity are deeply interconnected with each other and can influence and be influenced by tumor metabolism.

**Figure 4 cancers-11-01574-f004:**
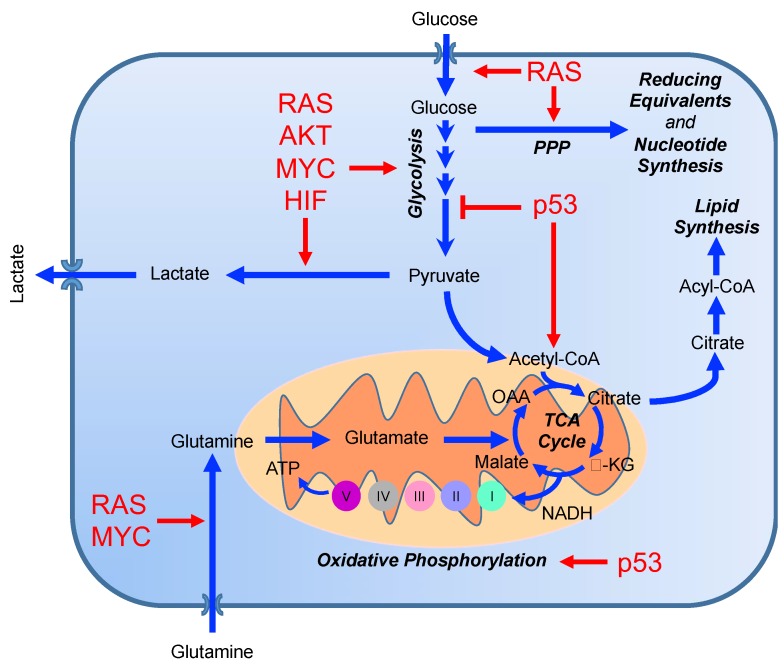
Effects of prototypical oncogenes and tumor suppressor genes on major metabolic processes. (PPP, pentose phosphate pathway; TCA, tricarboxylic acid cycle; OAA, oxaloacetate; α-KG, alpha-ketoglutarate).

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
