# Peer review of "The Many Facets of Tumor Heterogeneity: Is Metabolism Lagging Behind?"

_cancers, 2019, doi:10.3390/cancers11101574_

Round 1

Reviewer 1 Report

General comments

The authors present an interesting and well-written review on tumor metabolism, emphasizing on the intratumoral heterogeneity and the different metabolic strategies that a tumor can employ to adapt to ever evolving environments or to therapeutic insults.

Despite the thoroughness and accuracy of the presented manuscript, authors should consider that the text could seem overwhelming and difficult to follow by naïve readers not familiarized with tumor metabolism. The manuscript would gain if authors introduce illustrations to help the interpretation of the text. Perhaps breaking the text in shorter chapters would also help the reading.

Content-wise, authors could consider adding more information in the subject of hypoxia, as the hypoxic environments that normally characterize tumors is having profound impact in the regulation of tumor metabolic routes.

Author Response

Reviewer #1

General comments. The authors present an interesting and well-written review on tumor metabolism, emphasizing on the intratumoral heterogeneity and the different metabolic strategies that a tumor can employ to adapt to ever evolving environments or to therapeutic insults.

Despite the thoroughness and accuracy of the presented manuscript, authors should consider that the text could seem overwhelming and difficult to follow by naïve readers not familiarized with tumor metabolism. The manuscript would gain if authors introduce illustrations to help the interpretation of the text. Perhaps breaking the text in shorter chapters would also help the reading.

Content-wise, authors could consider adding more information in the subject of hypoxia, as the hypoxic environments that normally characterize tumors is having profound impact in the regulation of tumor metabolic routes.

We thank the reviewer for thoughtful comments that helped us to improve this revised version. We agree that the previous version failed to include hypoxia as one of the major drivers of metabolic reprogramming in tumors. We now include new text describing the role of HIF1ain regulating the response of glycolytic metabolism to hypoxia and genomic mutations of TCA enzymes that lead to impaired dioxygenase activity. Please see text lines 158-171 on pages 5-6.

We are pleased to know the reviewer found our work thorough and accurate, as we were worried it may have been too descriptive. Although we did not change to the main text, we added two new figures as suggested by the reviewer. One figure depicts models of tumor evolution (Figure 2), and the second summarizes the effects of prototypical oncogenes and tumor suppressor genes on key metabolic pathways during tumorigenesis. Moreover, to better guide readers through this complicated field, we referred to five excellent review articles authored by groups that elegantly explain the complexity of tumor metabolism (38-43).

Reviewer 2 Report

This is a nicely written review depicting the historical developments in the research and understanding of intratumor heterogeneity and the contributions from genetics and metabolism to that.

1. Did the authors seek permission from the original publisher for the reproduction of figure 1? If the did not, they should do it to prevent any legal trouble for the MDPI.

Figure 2 about intratumor heterogeneity needs to be redrawn and most of the texts on the “Metabolism” and “Microenvironment” panel are not visible.

Author Response

Reviewer #2

This is a nicely written review depicting the historical developments in the research and understanding of intratumor heterogeneity and the contributions from genetics and metabolism to that.

Did the authors seek permission from the original publisher for the reproduction of figure 1? If the did not, they should do it to prevent any legal trouble for the MDPI. Figure 2 about intratumor heterogeneity needs to be redrawn and most of the texts on the “Metabolism” and “Microenvironment” panel are not visible.

We thank the reviewer for these suggestions. Regarding the reproduction of Muller’s original drawings in Figure 1, we will discuss this point further with the editor (it is possible the book is no longer subjected to copyright because it was published in 1838). We downloaded the images from ECHO - Cultural Heritage Online, which is an open access infrastructure based in Berlin that “works to bring together sources, research, and institutional partners in order to use the Internet as a collaborative research tool” (http://echo.mpiwg-berlin.mpg.de/home). This website represents a remarkable collection of knowledge, the life sciences book section is simply extraordinary. Regarding Figure 2 (current Figure 3) we followed the referee’s advice and redrew the figure to increase font size.

Reviewer 3 Report

This is an extremely well-written and timely review article that gives a historical overview of our understanding of tumor heterogeneity and links this to our modern understanding of cellular and oncogene heterogeneity in tumors and finally describes our initial efforts to understand metabolic heterogeneity in tumors. There are two areas the authors have chosen not to review that I believe could make this piece more comprehensive. The tumor metabolism group at University of Texas Southwestern has published several studies interrogating the activity of metabolic pathways in human tumors, in vivo (PMID 30146487, 25525878, 26853473, 28985563). In several of these studies, regional sampling of several regions of individual tumors revealed different preferred metabolic pathway utilization. Summarizing these results would support the authors case that metabolic heterogeneity exists in human tumors and needs to be better understand. There is an extensive radiology/nuclear medicine literature describing human tumor metabolic heterogeneity with respect to phenotypes such as hypoxia, glucose uptake and nucleotide uptake (for example, see review article 25421725). Including a description of imaing-defined heterogentity would further support the authors case. Minor comments: Line 219, metabolic instead of metabolism

Author Response

Reviewer #3

This is an extremely well-written and timely review article that gives a historical overview of our understanding of tumor heterogeneity and links this to our modern understanding of cellular and oncogene heterogeneity in tumors and finally describes our initial efforts to understand metabolic heterogeneity in tumors. There are two areas the authors have chosen not to review that I believe could make this piece more comprehensive. The tumor metabolism group at University of Texas Southwestern has published several studies interrogating the activity of metabolic pathways in human tumors, in vivo (PMID 30146487, 25525878, 26853473, 28985563). In several of these studies, regional sampling of several regions of individual tumors revealed different preferred metabolic pathway utilization. Summarizing these results would support the authors case that metabolic heterogeneity exists in human tumors and needs to be better understand. There is an extensive radiology/nuclear medicine literature describing human tumor metabolic heterogeneity with respect to phenotypes such as hypoxia, glucose uptake and nucleotide uptake (for example, see review article 25421725). Including a description of imaing-defined heterogentity would further support the authors case. Minor comments: Line 219, metabolic instead of metabolism. 

We thank the reviewer for the thoughtful advice that has helped us improve the quality of our review. The reviewer is absolutely right, we cannot discuss metabolic heterogeneity without including DeBerardinis’s work. Both suggested parts are included in the revised review (please see lines 335-372 on page 9).